# The Role of Non-Coding RNAs in the Regulation of the Proto-Oncogene MYC in Different Types of Cancer

**DOI:** 10.3390/biomedicines9080921

**Published:** 2021-07-30

**Authors:** Ekaterina Mikhailovna Stasevich, Matvey Mikhailovich Murashko, Lyudmila Sergeevna Zinevich, Denis Eriksonovich Demin, Anton Markovich Schwartz

**Affiliations:** 1Center for Precision Genome Editing and Genetic Technologies for Biomedicine, Engelhardt Institute of Molecular Biology, Russian Academy of Sciences, 119991 Moscow, Russia; stasevich.em@phystech.edu (E.M.S.); shvarts@eimb.ru (A.M.S.); 2Laboratories for the Transmission of Intracellular Signals in Normal and Pathological Conditions, Engelhardt Institute of Molecular Biology, Russian Academy of Sciences, 119991 Moscow, Russia; murashko.mm@phystech.edu; 3Koltsov Institute of Developmental Biology, Russian Academy of Sciences, 119334 Moscow, Russia; lzinevich@gmail.com; 4Department of Molecular and Biological Physics, Moscow Institute of Physics and Technology, 141701 Moscow, Russia

**Keywords:** MYC, miRNAs, lncRNA, circRNA, cancer

## Abstract

Alterations in the expression level of the MYC gene are often found in the cells of various malignant tumors. Overexpressed MYC has been shown to stimulate the main processes of oncogenesis: uncontrolled growth, unlimited cell divisions, avoidance of apoptosis and immune response, changes in cellular metabolism, genomic instability, metastasis, and angiogenesis. Thus, controlling the expression of MYC is considered as an approach for targeted cancer treatment. Since c-Myc is also a crucial regulator of many cellular processes in healthy cells, it is necessary to find ways for selective regulation of MYC expression in tumor cells. Many recent studies have demonstrated that non-coding RNAs play an important role in the regulation of the transcription and translation of this gene and some RNAs directly interact with the c-Myc protein, affecting its stability. In this review, we summarize current data on the regulation of MYC by various non-coding RNAs that can potentially be targeted in specific tumor types.

## 1. Introduction

The MYC family of proto-oncogenes consists of three genes—C-MYC, L-MYC, and N-MYC [1]. The name of the family was coined after the discovery of homology between the human gene C-MYC, overexpressed in various tumors, and the oncogene v-Myc, carried by the avian myelocytomatosis virus (myelocytomatosis) [2]. Subsequently, homologs of c-Myc were discovered for humans: N-Myc [3], and L-Myc [4]. This review focuses on the most studied proto-oncogene of this family, C-MYC (or simply MYC).

C-Myc is an extraordinary transcription factor, as it has been shown to affect the expression of up to 15% of all genes in the human body [5]. It controls the expression of genes involved in a wide range of cellular processes, such as transcription, translation, cell cycle [6,7], cell adhesion [8], and others. Along with other factors of the MYC family, C-Myc has an important role in mammalian embryogenesis, especially in the development of cartilage, the liver, the thymus, submandibular glands, and brown adipose tissue [9,10,11]. This factor is also crucial for the normal development and activation of various populations of lymphocytes [12,13,14]. To regulate transcription, c-Myc forms a heterodimer with the transcription factor Max. Together, they can bind to a conserved E-box sequence (CACGTG) to activate or enhance the transcription of various genes. Moreover, c-Myc can bind to other transcription factors, such as TBP [15] or ETS2 [16], to DNA methyltransferases, such as Dnmt3a [17], or to histone-modifying enzymes, for example, ASH2L [18]. Overexpression of the MYC gene in various types of tumors is associated with an unfavorable prognosis. In particular, this correlation has been shown for Burkitt’s lymphoma [19,20], small cell lung cancer [21], breast cancer [22], and colorectal cancer [23].

Cancer is one of the central topics of modern science and many distinctive features inherent in the development of human cancers are well known. It is characterized by a tendency for rapid uncontrolled growth, an unlimited number of cell divisions, evasion of apoptosis and the immune response, metastasis, abnormal cellular metabolism, genomic instability, and stimulation of angiogenesis [24] (Figure 1).

Disruption of the cell division mechanism is one of the main characteristics of malignant tumors. Many studies have demonstrated a correlation between the expression of the MYC gene and the rate of cell proliferation [25,26,27,28,29] (Figure 1). C-Myc controls the expression of a number of key cell cycle regulators by stimulating or suppressing the expression of certain miRNAs. Thus, an increase in the level of c-Myc activates the synthesis of a number of positive regulators of proliferation: cyclins D and E, cyclin-dependent kinases CDK4 and 6, negative regulators of cell division, an inhibitor of cyclin-dependent kinase 1B (CDKN1B), and retinoblastoma protein (RB1). C-Myc also suppresses the expression of cyclin-dependent kinase inhibitor 1A (CDKN1A) [30]. In addition, c-Myc activates the expression of the MINA53 gene (Myc-induced nuclear antigen 53), the product of which stimulates the rapid growth of human glioblastoma, leukemia, and stomach cancer cells [31,32,33]. MINA53 is able to participate in the activation of gene expression by regulating the methylation status of histone H3K9me3 and it is also involved in the AP-1 signaling pathway, which is closely related to cell proliferation [31]. The expression of the ID1 gene is also regulated by c-Myc. In breast [34] and lung cancer cells [35], ID1 has been demonstrated to increase the rate of cell growth by regulating the expression of cyclins. Moreover, high levels of c-Myc stimulate the expression of thymidylate synthase (TS), inosine monophosphate dehydrogenase 2 (IMPDH2), and phosphoribosyl pyrophosphate synthetase 2 (PRPS2) by binding to their regulatory sequences. The expression levels of TS, IMPDH2, and PRPS2 positively correlate with the synthesis of nucleotides. In melanoma cells, it has been shown that cell proliferation is linked with the expression of TS, IMPDH2, and PRPS2 [36].

Healthy somatic cells have a limited number of divisions, while cancer cells are characterized by the ability to replicate endlessly. C-Myc has been shown to activate the expression of telomerase reverse transcriptase protein (TERT), which lengthens telomeres, allowing cells to divide an unlimited number of times [37] (Figure 1).

C-Myc is also involved in the regulation of genomic stability, the disturbance of which is another characteristic feature of tumor development (Figure 1). MYC expression has been shown to be associated with the reduced expression of telomeric repeat-binding factor 2 (TRF2) [38,39]. Knockout of TRF2 genes leads to DNA damage on telomeres and chromosomal aberrations [40]. Overexpression of MYC can also cause an increase in the amount of reactive oxygen (ROS) in cells, which can damage DNA and increase the frequency of mutations [41]. This may happen due to the stimulation of expression of a number of mitochondrial genes and, in general, the process of mitochondrial biogenesis, as well as a negative effect on the level of the enzyme superoxide dismutase (SOD), which eliminates ROS [42,43]. At the same time, the high expression of MYC contributes to the survival of cells with multiple DNA damage by activating their repair systems. C-Myc has been shown to bind to the promoter region of the NBS1, KU70, RAD51, BRCA2, and RAD50 genes, which are involved in the repair of double-stranded breaks [44].

The resistance of cells to chemotherapeutic agents is a serious obstacle to the successful treatment of cancer patients. It has been found that an increased expression of the MYC gene is associated with the resistance of cells to a number of drugs. The effect of c-Myc on the expression of DNA repair factors is one of the possible reasons for the survivability of cells with a high level of this transcription factor during treatment with chemotherapeutic DNA-damaging drugs and short-wave radiation [45,46]. Thus, the suppression of MYC expression in lung cancer [47], melanoma [48], ovarian cancer [49], and bladder cancer cells [50] increases the sensitivity of the tumor to cisplatin. Additionally, in breast cancer cells resistant to the estrogen agonist tamoxifen, as well as to doxorubicin and paclitaxel, increased expression of the proto-oncogene MYC has been found [51].

Another important indicator of the development of cancer and a possible explanation for its resistance to chemotherapy is the evasion of cells from apoptosis. In this process, the role of c-Myc is ambiguous (Figure 1). C-Myc has been shown to regulate the expression of the prothymosin alpha (PTMA) gene, which is responsible for cell proliferation in many types of cancer. PTMA suppresses the expression of the BAX and BAD genes, which stimulate mitochondrial apoptosis. When MYC expression is suppressed or PTMA promoter mutations occur at the c-Myc binding site, the cells become sensitive to sorafenib [52]. Additionally, c-Myc suppresses the expression of a number of miRNAs, which control anti-apoptotic genes BCL-2 and MCL1 [30]. At the same time, the increased expression of the MYC gene in the cell activates the ARF–Mdm2–p53 tumor suppressor pathway, which leads to the activation of apoptosis and, as a result, the suppression of tumor growth [53]. In the case of faulty apoptosis mechanisms in the cell, for example, sustained mutations in p53 or ARF, overexpression of MDM2, and changes in the regulatory pathways of BCL-2 and NF-kB [54,55], high expression of MYC does not lead to the death of tumor cells. It is also worth mentioning that a high level of MYC expression increases the resistance of myeloma cells to bortezomib by activating the pentose phosphate pathway. C-Myc has been shown to interact with the long non-coding RNA PDIA3P which enhances binding to the promoter of the G6PD gene involved in this metabolic pathway [56].

In addition to the pentose phosphate pathway, c-Myc also affects other cellular metabolic pathways (Figure 1). For tumor cells, the so-called Warburg effect is typical, which causes a suppression of pyruvate oxidation in mitochondria and an increase in the intensity of glycolysis and, consequently, this stimulates lactate production. This alteration can be explained by the necessity for rapid proliferation during oxygen deficiency and the possible adaptation of cells to cytotoxic agents, the effect of which is associated with oxygen metabolism [57]. C-Myc has been shown to regulate the expression of the glucose transporter gene GLUT1 by binding to the E-box sequence in its promoter. C-Myc is also able to activate the expression of monocarboxylate transporters (MCT1 and MCT2), which are responsible for the transport of the main product of oxygen-free metabolism, lactate [58]. C-Myc stimulates the expression of most glycolytic enzymes genes, including hexokinase II (HK2), pyruvate kinase m2 (PKM2), enolase 1 (ENO1), glyceraldehyde-3-phosphate dehydrogenase (GAPDH), and lactate dehydrogenase A (LDHA) [59,60]. Additionally, c-Myc plays a significant role in changing the metabolism of a number of amino acids: glutamine, proline, and branched-chain amino acids by activating glutaminase (GLS), P5C reductase (P5CR), and branched-chain aminotransferase (BCAT) expression [60,61]. Alterations in metabolism lead to the release of lactate, succinate, and glutamine by tumor cells, which contribute to the attraction of macrophages to the tumor and their polarization into the immunosuppressive phenotype M2 [62].

Active growth of solid tumors is impossible without interaction with the microenvironment, in particular, without the ability to avoid an immune response. In a series of experiments, it was shown that c-Myc regulates the expression of immune checkpoints in cells [63] (Figure 1). Suppression of c-Myc reduces the expression of the innate immunity regulator CD47 and the adaptive immunity checkpoint PD-L1, thus enhancing the antitumor immune response [64]. CD47 is known to interact with the signaling regulatory protein α SIRPa on the macrophage surface, preventing phagocytosis of the body’s cells by macrophages. The PD-L1 ligand interacts with the PD-1 receptor on the surface of T-lymphocytes, suppressing their activity [65]. Moreover, some data suggest that Myc is able to down-regulate the expression of HLA class I in various cancers [66].

Tumor metastasis is another hallmark of cancer development. The cell undergoes an epithelial–mesenchymal transition in order for the cancer cells to spread and further consolidate in different parts of the body. Many experimental groups have shown a link between c-Myc regulation and cellular metastasis [67] (Figure 1). In clear cell renal cell carcinoma, PIM1-mediated phosphorylation of c-Myc activates transcription factors ZEB1, ZEB2, Snail1, Snail2, and Twist, which further trigger the epithelial–mesenchymal transition program and increase the likelihood of tumor metastasis [68]. Additionally, c-Myc promotes expression of miR-9-5p which controls the leukemia inhibitory factor receptor (LIFR) and suppressor of cytokine signaling 5 (SOCS5). LIFR inhibits metastasis through the Hippo/YAP pathway, and SOCS5 inhibits cell migration by inhibiting the JAK/STAT pathway [30,69,70]. A correlation between MYC expression and metastasis was also shown in non-small cell lung cancer [71], breast cancer [72], and gallbladder cancer [73].

For effective growth, cancer cells activate the mechanisms of angiogenesis (Figure 1). In the normal state, cells that do not receive enough oxygen can induce the expression of VEGF, which stimulates the development of new blood vessels. This mechanism is often used by cancer cells to vascularize the tumor. It was found that in leukemia cells, c-Myc binds to the VEGFA promoter sequence and thereby increases its expression [74]. The same mechanism of c-Myc’s influence on VEGFC expression was found in cells of pancreatic neuroendocrine tumors [75]. The promoting effect of c-Myc on the expression of VEGF family factors was also shown in non-small cell lung cancer [71]. Moreover, c-Myc stimulates the expression of miRNAs that control the synthesis of a number of angiogenesis inhibitors: members of the TGF-β signaling pathway (TGF beta receptor 2 (TGFBR2) and mothers against decapentaplegic homolog 4 (SMAD4)), thrombospondin 1 (THBS1), and connective tissue growth factor (CTGF) [30,76,77].

As can be seen from the above data, c-Myc is involved in almost all mechanisms of oncogenesis of various types of tumors. At the same time, it should be noted that a small change in MYC expression (sometimes less than two-fold) is often enough to change the processes of oncogenesis [78,79,80]. For effective and long-term suppression of the expression of this proto-oncogene, it is necessary to know in detail the mechanisms that control the transcription of this gene, the stability of its mRNA and its translation, as well as the factors responsible for the stability of the Myc factor itself [25,46,81] (Figure 2).

The bromodomain-containing protein 4 (BRD4) is a universal transcription regulator which also controls the transcription of the MYC proto-oncogene (Figure 2). Inhibition of BRD4 by thienotriazolodiazepine JQ1 in colorectal cancer cells reduces MYC expression and inhibits cell proliferation [82]. A similar effect is observed in retinoblastoma cells, where BRD4 inhibition induces cell cycle arrest and apoptosis [83]. In neuroblastoma, lung carcinoma, colon adenocarcinoma, and melanoma cells, dual PI3K/BRD4 inhibition by SF2523 contributes to a decrease in c-Myc levels and markedly inhibits the growth and metastasis of cancer cells [84,85]. Another bromodomain-containing protein, bromodomain PHD transcription factor (BPTF), can activate MYC expression. It has been shown that suppression of BPTF transcription and the use of BPTF inhibitors lead to a decrease in the expression of the MYC gene [86,87].

Proteins that interact directly with the c-Myc protein can also affect its gene transcription. In lung and breast cancer cells, a correlation was shown between the expression of the ZNF121 and MYC genes: during the siRNA-mediated knockdown of ZNF121, MYC expression decreased and, accordingly, when ZNF121 was overexpressed, MYC expression increased [88,89] (Figure 2). Among other things, suppression of the ZNF121 gene reduced the rate of proliferation in breast cancer cells [89].

It has been shown that in human fibroblasts, FOXO3a binds to the region in the c-MYC promoter, and this interaction activates the transcription of the c-MYC gene [90]. On the other hand, the interaction of the promoter of this gene with the proteins of the SMAD family leads to the suppression of expression of the MYC gene [91,92].

IGF2BP1/2/3 (mRNA-binding proteins of insulin-like growth factor 2) are able to bind to many mRNAs, including the c-Myc mRNA, recognizing the GG(m6A)C sequence, and by this binding, it stabilizes the mRNA. It has been shown that suppression of IGF2BP1/2/3 expression in cervical cancer and liver cancer cells leads to a decrease in the amount of c-Myc protein, as well as to a decrease in the rate of proliferation [93] (Figure 2).

AU-rich element RNA-binding protein 1 (AUF1) binds to AU-rich mRNA regions and triggers the mRNA degradation process. It has been shown that the suppression of AUF1 does not lead to a change in the level of MYC mRNA, but reduces the amount of c-Myc protein in cells, which suggests that AUF1 may affect the translation of this mRNA (Figure 2). In addition, suppression of AUF1 led to a decrease in the rate of proliferation in leukemia, colon cancer, and cervical cancer cells [94,95].

C-Myc is a short-lived protein, so the mechanisms responsible for its stability and degradation play an important role in tumor development. In tumors with a high level of c-Myc, improper functioning of the mechanisms of its ubiquitination can be observed. It is important to note that different types of ubiquitin ligases have different effects on the stability of this transcription factor (Figure 2). For example, ubiquitin ligase FBXW7 and E3 ubiquitin ligase adapter SPOP promote the degradation of c-Myc [96,97], while ubiquitin ligases SKP2 and HUWE1, on the contrary, improve the stability of this protein. In multiple myeloma, suppression of HUWE1 expression leads to a decrease in c-Myc levels and inhibition of tumor growth [98]. Enzymes deubiquitinating c-Myc have also been shown to affect its stability. Thus, suppression of USP28 and USP36 reduces the c-Myc level and suppresses cell proliferation [99,100]. Glycosyltransferase OGT has been shown to enhance cell proliferation by stabilizing the c-Myc protein by combining it with β-N-acetylglucosamine [101]. Increased OGT expression was found in many tumors, including prostate [102], breast [103], lung, and colon cancers [104]. Lowering the level of OGT mRNA leads to a decrease in c-Myc protein in prostate cancer cells [102]. Another protein, cancer inhibitor of protein phosphatase 2A (CIP2A), has increased expression levels in colorectal cancer [105], stomach cancer [106], prostate cancer [107], and multiple myeloma [108]. CIP2A has been shown to prevent the degradation of the c-Myc protein by inhibiting the activity of phosphatase PP2A. Phosphatase PP2A dephosphorylates c-Myc at serine 62, which is necessary for ubiquitination by ubiquitin ligase FBXW7 and initiation of degradation [109] (Figure 2).

A more detailed understanding of the regulation of MYC expression in cancer cells opens up new targets for drug discovery and new approaches in the treatment of cancer. Recently, many groups of scientists have confirmed that non-coding RNAs play an important role in regulating cellular processes, blocking or activating the transcription and translation of this gene, or interacting with the c-Myc protein directly. In tumor cells, shifts in the expression of many non-coding RNAs may be involved in tumor development [110,111]. It is important that the expression of some RNAs is specific to certain types of cancer. This makes non-coding RNAs a convenient target for suppressing tumor development with minimal possible impact on healthy cells [112]. Among other things, non-coding RNAs have shown themselves to be a promising marker for the diagnosis of oncogenic diseases [113]. This diagnostic method is convenient, as non-coding RNAs can be easily detected in the cells and biological fluids of the patient. For example, the detection of lncRNA PCA3 in urine is widely used as a marker of prostate cancer [114]. Similarly, the lncRNA AA174084 in gastric juice is a potential biomarker for the early diagnosis of gastric cancer [115].

In this review, we will examine in more detail the effect of miRNAs, long non-coding RNAs, and circular RNAs on the expression of the MYC proto-oncogene in various types of cancer.

## 2. miRNAs

MicroRNAs (miRNAs) are a class of small, endogenous, single-stranded non-coding RNA molecules. They act as a sequence-specific tool that is widely used in nature to regulate gene expression. At the moment, several dozens of miRNA variants that affect the expression of the MYC gene have been analyzed. Most of these miRNAs bind directly to the mRNA of the MYC gene [30]. Others affect its level by regulating genes that control the stability of the c-Myc protein. For example, miR-375-3p suppresses the expression of the CIP2A gene, the product of which is involved in the stabilization of c-Myc due to the phosphorylation of Ser62 [116]. Another example in mouse hepatoma cells is miR-24 regulating the OGT gene that increases the stability of the c-Myc protein by combining with β-N-acetylglucosamine [117].

As described above, high expression of the MYC gene is characteristic of many types of cancers. In this regard, it is not surprising that in tumors, the levels of most miRNAs that control the expression of the MYC gene are often reduced. The possibility of using miRNA complementary to the MYC gene sequence is being considered as a targeted therapy for cancer [118,119]. The use of miRNA leads to a reduced survival rate of tumor cells of different types of cancer, suppression of their reproduction, and migration [120,121,122,123]. It is important to note that for some miRNAs that bind to the mRNA of the MYC gene, a protective effect for tumor cells was also revealed. Thus, it was shown that Hodgkin’s lymphoma cells can have a high level of miR-24-3p, which limits the expression of CDKN1B/P27kip1 and MYC genes and also protects cells from apoptosis [124]. On the other hand, a reduced level of miR-24-3p is observed in breast cancer and nasopharyngeal carcinoma cells, increasing the metastatic potential of tumor cells [125,126]. In another study, it was found that hepatocellular carcinoma cells with a lower level of miR-17-5p have greater metastatic activity, but a lower survival rate compared to cells of this tumor with more highly expressed miRNA [127]. Several studies have shown that the expression of many miRNAs differs significantly both in normal human tissues and in different types of tumors [128,129]. Thus, to study the possibility of using miRNA in therapy, it is necessary to take into account which RNAs control c-Myc levels in different types of cancer and their mechanisms.

### 2.1. miRNAs Controlling the Expression of the MYC Gene in Various Types of Cancer

For various types of cancer, the role of miRNAs of the let-7 family in regulating the expression of the MYC gene has been described (Table 1; Figure 3). A decreased level of these miRNAs in tumor cells is associated with a negative prognosis in patients with acute myeloid leukemia [130], breast cancer [131], stomach cancer [132], liver cancer [133], and neuroblastoma [134]. The expression level of let-7 is inversely correlated with the metastatic activity of prostate cancer [135]. Overexpression of miRNA of the let-7 family leads to the suppression of the proliferation of cells of breast [136,137,138], liver [139,140], lung [141], and colon cancers [142,143], and B-cell lymphomas [144,145,146]. The cancer-fighting qualities of the let-7 miRNA family are also explained by their effect on the expression of other proto-oncogenes: K-RAS, HMGA2, and cyclin D1 and D2 [147]. However, in some cases, members of let-7 miRNA family can stimulate the development of a tumor, for example, a high level of let-7g stimulates the progression of osteosarcoma [148].

The expression, stability, and activity of miRNAs of the let-7 family are regulated by various factors. The most interesting is the regulatory loop with the MYC gene. An increase in the level of c-Myc boosts the expression of the LIN28A and LIN28B genes, whose products trigger the degradation of the let-7 family miRNAs [203] (Figure 3). Thus, artificially expressed miRNAs can lead to stimulation of the endogenous production of let-7 by suppressing the MYC expression. It has also been shown that the level of miRNAs of this family increases in breast cancer cells in response to estrogen. It is assumed that this effect serves to limit the stimulation of MYC expression by the same hormone [136]. Another way to regulate miRNA activity is to inactivate them by binding to long non-coding RNAs, so-called competing endogenous RNAs (ceRNA). Thus, ceRNA H19 couples with let-7b in breast cancer cells, activating epithelial–mesenchymal transition processes [137], and CCAT1 RNA binds miRNA of the let-7 family in hepatocellular carcinoma cells, stimulating their proliferation and migration [140]. The role of ceRNA will be described more precisely in the next section.

MiR-34 is another miRNA family that controls the expression of the MYC gene in various tumor types (Table 1). Reduced expression of miRNAs of this family in tumor cells is associated with increased metastatic activity in patients with prostate cancer [194], as well as breast, lung, and colon cancers, melanoma, and head and neck tumors [153]. An artificial increase in the expression of these miRNAs leads to suppression of the proliferation of gastric [163] and prostate [192] cancers, head and neck tumors [173], and B-cell lymphoma [204] and also suppresses the tumor transformation of kidney epithelial cells [199]. It has been shown that the tissue-specific factor gastrokine-1 stimulates the expression of miR-34a in gastric cancer cells, suppressing the expression of proto-oncogenes MYC (Figure 3) and RhoA, which leads to a decrease in the ability of cells to migrate and invade [163]. Stimulation of miRNA miR-34a expression occurs when the tumor repressor p53 is activated [184]. Activation of p53 also leads to an increase in the expression of another miRNA, miR-145-5p, which also controls the expression of the MYC gene [154] (Figure 3). These data demonstrate that the stimulation of the expression of miR-34a and miR-145-5p is significant in the antitumor activity of p53 in various types of cancer. Overexpression of miR-145-5p considerably suppresses the proliferation of breast and colon cancer cells [154], lung cancer cells [181], prostate cancer cells [193], gastric cancer [167], and oral squamous carcinoma cells [189].

### 2.2. miRNAs That Control the Expression of the MYC Gene in Breast Cancer Cells

The influence of miRNAs of other families on the expression of the MYC gene has been shown in certain types of tumors. Thus, for breast cancer, in addition to the previously described let-7, miR-34, miR-145-5p, and miR-24-3p, a contribution to the regulation of MYC expression was shown for several other miRNAs with more distinct tissue specificity (Table 1). For example, in addition to let-7, two other miRNAs that control the expression of the MYC gene are involved in coordinating the response to estrogen in breast cancer cells: miR-21-5p and miR-98-5p [136]. The expression of miRNAs miR-17-5p and miR-20a-5p that suppress the MYC gene is activated in breast cancer cells by the c-Myc factor, which demonstrates their participation in the negative regulation of the expression of this factor [152] (Figure 3).

### 2.3. miRNAs That Control the Expression of the MYC Gene in the Cells of Tumors of the Digestive System

For gastric cancer, many miRNAs have been found that control the expression of the MYC gene (Table 1). In addition to the previously described let-7, miR-145, and miR-34, the expression of this proto-oncogene is controlled by miRNAs miR-212-3p, miR-429, miR-125, miR-494-3p, miR-155-5p, miR-33b-5p, miR-25-5p, miR-150-5p, miR-1304, miR-590-3p, miR-449c-5p, and miR-561-3p [115,122,152,160,161,162,164,165,166,168,169]. In tumor cells, the levels of most of these miRNAs, with the exception of miR-25-5p, are lower than in normal tissue, and artificially increasing their expression leads to suppression of tumor cell proliferation and their ability to invade neighboring tissues. On the contrary, miR-25-5p RNA is hyperexpressed in gastric adenocarcinoma cells compared to normal tissue. Increased expression of this RNA is associated with a higher survival rate of cancer cells [162] (Figure 3).

MiR-33b and miR-93 have been shown to reduce MYC expression in bowel cancer cells (Table 1). Suppression of the activity of these miRNAs leads to an increase in the ability of the tumor to grow and form metastases [155]. Four other miRNAs that control MYC expression were also found in cells of this type of cancer: miR-200b-3p, miR-182-5p, miR-182a-5p, and miR-320b (Figure 3). The expression of all these RNAs is reduced in tumor cells, and their overexpression suppresses the proliferation of rectal cancer cells [156,157,158].

Regulation of MYC gene expression by miRNAs of the miR-320 family has been shown for liver cancer cells. Increased expression of miR-320a inhibits the ability of hepatocellular carcinoma cells to grow invasively [177]. MiRNA let-7, miR-148a-5p, miR-363-3p, miR-744-5p, miR-599, miR-9, miR-185-5p, miR-526b, miR-17-5p, and miR-122-5 are also involved in regulating the expression of the MYC proto-oncogene in liver cancer cells (Table 1). Constitutive overexpression of these miRNA suppresses the proliferation of cancer cells and their ability to invade [127,133,140,174,175,176,178,179,180]. For three of these RNAs, miR-148a-5p, miR-363-3p, and miR-122-5, negative feedback was shown with the expression of the MYC gene (Figure 3). Thus, c-Myc has been shown to directly inhibit the activity of these RNA promoters in liver cancer cells [174,179]. It is worth noting that unlike miR-148a-5p and miR-122-5, which directly interacts with the mRNA of the MYC gene, miR-363-3p suppresses the expression of ubiquitin-specific protease 28, that stabilizes the c-Myc protein [174]. For miRNAs miR-17-5p, miR-9, and miR-185-5p, positive feedback was shown with MYC gene expression; transcription factor c-Myc stimulates transcription of these miRNAs in liver cancer cells [133,178] (Figure 3). Interestingly, in contrast to miR-17-5p, the expression levels of miR-9 and miR-185-5p in tumor cells are higher than in normal tissues [178]. A high level of miRNA suppressing MYC expression can be combined with a high level of transcription of this proto-oncogene in tumor cells due to ceRNA, which binds and inactivates certain miRNAs. Therefore, earlier in the liver cells, an increased level of RNA CCAT1, which binds to miRNA of the let-7 family, was detected [140]. A specific ceRNA, Linc00176, was also found for miRNAs miR-9 and miR-185-5p. Its enhanced expression level disrupts the reverse regulation of MYC gene expression in hepatocellular carcinoma cells, creating conditions for consistently high MYC expression. For this reason, this ceRNA can be considered as an important target for the development of therapy [178].

### 2.4. miRNAs That Control MYC Gene Expression in Lung Cancer Cells

An intriguing study was devoted to the negative effect of cigarette smoke on the expression of miR-487b-3p in lung cancer cells. This RNA suppresses the expression of a number of proto-oncogenes, including MYC, and its constitutive expression leads to a decrease in the proliferation and ability of lung cancer cells to invade [182]. In addition to the RNA families let-7, miR-34, and miR-145 mentioned in other sections, the expression of the MYC gene in lung cancer cells is also controlled by miR-199a-5p, miR-449c-5p, and miR-451a (Table 1). As expected, the expression levels of these RNAs in tumor cells are lower than in normal tissue, and a constitutive increase in their expression level leads to impaired proliferation and mesenchymal–epithelial transition of tumor cells [146,183,185]. Some miRNAs affect the expression level of the MYC gene by affecting the mRNA of factors that regulate the transcription of this oncogene. Thus, miR-4302 interacts with ZNF121 mRNA, lowering the level of the factor that activates the transcription of the MYC gene. The binding of this RNA by circRNA-103809 in lung cancer cells leads to an increase in the ability of the tumor for invasive growth [88] (Figure 3).

### 2.5. miRNAs That Control the Expression of the MYC Gene in Prostate Cancer Cells

In addition to the RNA families let-7, miR-34, and miR-145 mentioned before, the expression of the MYC gene in prostate cancer cells is controlled by miR-3667-3p and miR-33b (Table 1). The expression of the latter in tumor cells is suppressed by the cullin-4B protein, the mutation of which is characteristic of different cancer types [195,197] (Figure 3). Recently, it has also been found that the expression of miR-449a in prostate cancer cells increases in response to ionizing radiation at a dose of 4–8 Gy and, by suppressing the expression of the MYC gene, increases the sensitivity of these cells to radiation. Increasing the expression of such RNAs can be used to enhance the effectiveness of tumor radiotherapy [196]. In prostate cancer cells, dysregulation of MYC expression was also found due to an increased level of ceRNA MYU, which is able to bind to miRNA miR-184 [198] (Figure 3). The same miRNA is involved in the regulation of c-Myc levels in nasopharyngeal cancer cells. MiR-184 has been shown to inhibit MYC expression and tumor cell proliferation in response to increased levels of the tumor suppressor PDCD4 [188].

### 2.6. miRNAs That Control MYC Gene Expression in Blood Cancer Cells

Besides the RNA families let-7 and miR-34, the expression of the MYC gene in Burkitt lymphoma cells is controlled by miR-132-5p, miR-125b-1, miR-154, and mir-98 (Table 1). The expression of these miRNAs is suppressed in tumor cells, and their constitutive expression inhibits the proliferation of lymphoma cells [120,121]. In other types of blood cancers, specific miRNAs involved in the regulation of MYC gene expression have also been discovered. For example, a low level of the miRNAs miR-451a and miR-709 has been shown to have an important role in the development of acute T-cell leukemia [200]. Suppression of the expression of two other miRNAs that control the level of the MYC proto-oncogene, miR-126-5p and miR-29a-3p, is necessary for the survival and reproduction of myeloma cells [187,205] (Table 1). The expression of miR-126-5p in myeloma cells is suppressed by histone methyltransferase MMSET, the level of which can be increased in tumor cells as a result of a translocation between chromosomes 4 and 14 [187] (Figure 3). Additionally, in acute myeloma cells, it has been revealed that the lncRNA CCAT1 binds to miR-155, which leads to an increase in the level of MYC expression [149]. The use of these miRNAs and their analogs for tumor therapy is not yet common practice, but the level of expression of some miRNAs can be regulated using low-molecular-weight substances. For example, PRIMA-1Met causes an increase in the expression of miR-29a-3p in multiple myeloma cells, which leads to a decrease in the level of c-Myc and reduces the survival rate of tumor cells [205] (Figure 3).

### 2.7. miRNAs That Control the Expression of the MYC Gene in the Cells of Tumors of the Nervous System

In glioma cells, the level of c-Myc is controlled by miR-29b-1, the expression of which is suppressed by neurotensin (Figure 3). Decreased expression of the neurotensin receptor restored the level of this miRNA and suppressed the proliferation of tumor cells [170]. In patients with glioma, there is an inverse correlation between survivability and the expression of another RNA, miR-135a-5p, which suppresses the expression of the MYC gene [172] (Table 1). In the cancers of the nervous system, glioma and medulloblastoma, miRNA miR-33b-5p disturbs the regulation of MYC expression [171,186]. When searching for small molecules as anti-cancer drugs, it was found that lovastatin can increase the expression of miR-33b-5p in medulloblastoma cells [186] (Figure 3).

### 2.8. miRNAs That Control the Expression of the MYC Gene in Thyroid Tumor Cells

Another RNA of the miR-33a family, miR-33a-5p, is involved in the regulation of MYC expression in thyroid cancer. Suppression of the expression of this miRNA may be associated with the activity of the XB130 protein, and inhibition of this factor led to stunted growth of tumor cells [202] (Figure 3).

## 3. lncRNA

Long non-coding RNAs can control the level of active factor c-Myc at different levels: (1) at the level of transcription, by attracting transcription factors to the MYC gene regulatory sequence; (2) at the level of mRNA stability of this gene, by recruiting specific miRNA; (3) at the level of protein stability and by regulating the efficiency of c-Myc binding to DNA regulatory sequences (Figure 4). Several lncRNAs have been shown to be involved in regulatory loops associated with MYC gene expression in different tumor types. For example, the lncRNA c-Myc inhibitory factor (MIF), found in B-cell lymphoma cells, is synthesized with the participation of c-Myc factor, but by binding miR-586 it activates the expression of ubiquitin ligase E3, which promotes the degradation of c-Myc factor. Increased expression of MIF lncRNA suppresses the proliferation of lung cancer and cervical cancer cells [59]. Another lncRNA involved in a regulatory loop with the MYC gene is the ovarian adenocarcinoma-amplified lncRNA OVAAL. This lncRNA stimulates the activity of the MAPK cascade, including ERK kinase, which stabilizes c-Myc factor by phosphorylating it at serine 62. OVAAL RNA expression, in turn, is stimulated by c-Myc factor. Increased levels of this lncRNA promote the survival and proliferation of melanoma and colon cancer cells [206]. The expression of an antisense lncRNA of glutaminase (GLS-AS) can be suppressed in some tumor types, and this correlates with high levels of glutaminase. This enzyme can interact with c-Myc, increasing its stability. Interestingly, c-Myc factor itself suppresses GLS-AS expression [207] (Figure 4).

### 3.1. lncRNAs Controlling MYC Gene Expression in Different Tumor Types

Several lncRNAs are currently known to regulate MYC gene expression in various tumor types (Table 2). Of these, the most studied is the ceRNA colon cancer-associated transcript-1 (CCAT1) whose increased expression was first detected in colon cancer cells in 2011 [208]. This lncRNA has been shown to stimulate tumor growth, vascularization, and metastatic activity [209]. Increased expression of CCAT1 was also found in leukemia, lung, gastric, liver, gallbladder, kidney, prostate, and ovarian cancer cells. CCAT1 lncRNA stimulated cell survival, proliferation, and migration in these tumors [149,208,210,211,212,213,214,215]. Thus far, two main mechanisms of action of this RNA on MYC gene expression are known. Firstly, CCAT1 is involved in the spatial proximity of its locus (MYC-515), located 515 kb before the MYC promoter, and the enhancer (MYC-335), located 335 kb before the aforementioned promoter. This interaction enhances the transcription of this proto-oncogene in tumor cells [216]. Secondly, as mentioned in the previous section, CCAT1 protects the MYC gene mRNA by binding miRNAs let-7 and miR-155 [140,149] (Figure 4).

Another lncRNA from the same family, CCAT2, also increases c-Myc levels in colon cancer cells, but by recruiting the transcription factor TCF7L2 to the MYC gene promoter [226]. It was shown that the expression level of this lncRNA in ovarian cancer cells can be suppressed by vitamin D metabolites, which reduces the ability of tumor cells for invasive growth [253] (Figure 4). Additionally, high levels of CCAT2 lncRNA enhance the ability of osteosarcoma and hepatocellular carcinoma cells to invade and proliferate [241,250] and improve the resistance to radiotherapy of esophageal cancer cells [248]. The more common rs6983267(G) polymorphism variant of the CCAT2 gene has been shown to be associated with increased MYC gene expression levels and accelerated cervical cancer progression [260].

Another lncRNA affecting MYC expression in various tumor types is NEAT1, which forms specific nuclear structures called paraspeckles. These structures are involved in the maturation and retention of different types of RNA in the nucleus [261]. Elevated NEAT1 levels are associated with suppression of miR-34b activity and increased MYC gene expression in B-cell lymphoma cells [159]. In addition, NEAT1 is involved in the activation of histone acetylation in the MYC gene promoter region, activating its function [233]. It is worth noting that NEAT1 expression is in turn repressed by the c-Myc factor, which creates a negative regulatory loop [159] (Figure 4). Constitutive repression of NEAT1 lncRNA expression decreases proliferation capacity, reduces survival, and increases chemotherapeutic drug sensitivity in chronic myeloid leukemia [225], diffuse B-cell lymphoma [159], bladder cancer [219], uterine cancer [236], and rectal cancer [233,234].

Another lncRNA whose expression correlates positively with MYC expression is THOR. This lncRNA interacts with the insulin-like growth factor 2 mRNA-binding protein (IGF2BP1). The THOR–IGF2BP1 complex increases the mRNA stability of several proto-oncogenes, including the MYC gene [246] (Figure 4). Suppression of this lncRNA’s expression leads to decreased proliferation and migration ability of colon cancer cells [228]. High THOR expression accelerates tumor transformation of retinoblastoma cells [258] and growth of osteosarcoma, nasopharyngeal, and renal tumors [247,251,257].

GHET1 lncRNA also increases the stability of MYC gene mRNA through interaction with IGF2BP1 protein (Figure 4). Suppression of this lncRNA expression in gastric and colorectal cancer cells leads to reduced c-Myc levels and suppression of tumor cell proliferation [230,262]. High levels of GHET1 lncRNA expression in tumor cells are associated with poor prognosis in patients with lung, breast, head and neck, nasopharyngeal, stomach, liver, pancreatic, bowel, bladder, and osteosarcoma cancers [218]. High levels of expression of LINRIS lncRNA have been detected in colon cancer cells. This lncRNA stabilizes IGF2BP2, another member of this family of proteins, that extend the lifespan of MYC mRNA [232] (Figure 4).

Amplification of the locus containing the MYC gene has been observed in many tumor types. Moreover, the same locus contains several genes encoding lncRNAs. The expression of one such lncRNA, PVT1a, was shown to be up-regulated in 98% of tumors with amplification of the locus containing the MYC gene. Moreover, suppression of this lncRNA expression in such cells resulted in reduced MYC expression levels and suppressed proliferation [227]. It was found that PVT1a lncRNA can interact with the c-Myc factor, preventing its degradation. Suppression of this lncRNA’s expression has been shown to reduce the ability of lung, colon, and bladder cancer cells to proliferate, migrate, and grow invasively [220,227,244,263]. Recently, it was also shown that PVT1a lncRNA stimulates invasive growth of hepatitis B virus-infected liver cancer cells through stimulation of MYC gene transcription; this lncRNA blocks histone methyltransferase EZH2, which inhibits MYC promoter activity through methylation of lysine 27 on histone H3 [240] (Figure 4).

While searching for potentially oncogenic lncRNAs, EPIC1 RNA was found. This lncRNA interacts directly with the c-Myc protein and stimulates binding of this transcription factor to the promoters of genes controlling the cell cycle. It has also been shown that lncRNA EPIC1 can moderately enhance the Myc–Max interaction [221] (Figure 4). In addition to binding to the c-Myc factor, EPIC1 lncRNA is a potential regulator of the AKT-mTORC1 signaling pathway. The mTOR-specific inhibitor rapamycin is used for the therapy of some types of cancer, but cases of resistance to this drug have been described [264]. EPIC1 knockdown makes resistant breast and ovarian cancer cells sensitive to rapamycin [252]. High expression of EPIC1 lncRNA accelerates proliferation of lung cancer cells [243] and cholangiocarcinoma cells [224] and enhances invasive growth of colon cancer cells [229].

### 3.2. lncRNAs Controlling MYC Gene Expression in Digestive Tumors

Several other lncRNAs controlling the level and stability of c-Myc factor in tumor cells were found for digestive system cancers (Table 2). For example, Linc-RoR lncRNA stabilizes MYC gene mRNA in colon cancer cells by controlling its interaction with AU-rich element RNA-binding protein 1 (AUF1) and heterogeneous nuclear ribonucleoprotein I (hnRNPI) [223]. The expression of this lncRNA was also elevated in esophageal tumors [249]. In oral squamous cell cancer cells, Linc-RoR lncRNA binds miRNA miR-145-5p, blocking its binding to MYC gene mRNA [190] (Figure 4). A similar mechanism has been described for other lncRNAs whose increased expression is associated with high levels of c-Myc in cancer cells of the digestive system. For example, in gastric cancer cells, the ceRNA HOXC-AS1 binds miR-590-3p [165], in colon cancer cells the ceRNA SNHG3 suppresses miR-182-5p activity [157], and ceRNA Linc00176 blocks the binding of miR-9 and miR-185-5p to MYC mRNA in hepatocellular carcinoma cells [178] (Figure 4). Enhanced expression of CMPK2 lncRNA, which stabilizes far upstream element (FUSE)-binding protein 3 (FUBP3) and promotes its binding to the MYC gene regulatory element, was also found in colon cancer cells, resulting in activation of transcription of the MYC proto-oncogene [235]. Under conditions of glucose deficiency in rectal cancer cells, GLCC1 lncRNA expression is activated, which activates the interaction of the transcription factor c-Myc with the heat shock protein Hsp90, which prevents ubiquitination and degradation of this factor [231] (Figure 4).

### 3.3. lncRNAs Controlling MYC Gene Expression in Urinary Tumor Cells

Increased expression of GClnc1 lncRNA is an indicator of lower survival chances in bladder cancer. High levels of GClnc1 significantly promoted cell proliferation, metastasis, and tumor invasiveness [217]. GClnc1 binds to LIN28B and activates this protein, and LIN28B, as described in the previous section, is involved in degrading the miRNA of the the miR-let-7 family that controls MYC gene expression (Figure 4).

For lncRNA FILNC1, the ability to bind to the previously mentioned AUF1 protein, which controls the stability of many cellular mRNAs, including MYC, was shown (Figure 4). FILNC1 lncRNA expression in renal cancer cells is stimulated under conditions of ATP deficiency and leads to suppression of MYC expression and decreased tumor cell survival. Low levels of FILNC1 lncRNA in renal tumor cells are associated with a negative prognosis [256].

### 3.4. lncRNAs Controlling MYC Gene Expression in Prostate Cancer Cells

An interesting mechanism for regulating MYC oncogene expression was found in prostate cancer cell culture by switching the expression of three overlapping lncRNAs, NAT6531, NAT6558, and NAT7281. The scenario in the cell in this case is determined by the work of histone deacetylases. Their high activity promotes the transcription of only NAT6531 lncRNA. This lncRNA is a substrate for DICER nuclease, which slices it to form small RNAs that bind to MYC gene RNA and act as miRNA (Figure 4). Weak suppression of deacetylase activity increases the acetylation of histone H3 at the locus described, which blocks the transcription of NAT6531 and activates the transcription of lncRNA NAT6558. NAT6558 lncRNA does not form a loop that interacts with DICER nuclease and is not a source of small RNAs that decrease the half-life of MYC gene mRNA. When histone deacetylases are completely repressed, the longest lncRNA of this group, NAT7281, is synthesized and the transcription of NAT6531 and NAT6558 is blocked. Expression of NAT7281 leads to a strong suppression of MYC gene transcription [254] (Figure 4). Another lncRNA has been shown to be involved in the regulation of MYC gene expression in prostate cancer cells. PCGEM1 is a prostate-specific lncRNA that is up-regulated in various tumors of this organ and stimulated by androgens. This lncRNA interacts directly with the promoter region of the MYC gene, stimulating its transcription [255]. MYC expression was also found to be up-regulated in prostate cancer cells by elevated levels of ceRNA MYU which binds miR-184. Suppression of MYU RNA expression resulted in decreased levels of MYC expression and suppression of tumor cell proliferation [198]. Another ceRNA found in prostate tumors, PCAT-1, binds miR-3667-3p (Figure 4). Suppression of expression of this ceRNA results in reduced MYC expression and suppression of cancer cell proliferation [195].

### 3.5. lncRNAs That Control MYC Gene Expression in Breast Cancer Cells

Increased expression of LINC01638 lncRNA has been detected in breast cancer tissues compared to normal tissue. This lncRNA promotes the proliferation of breast cancer cells with a triple-negative phenotype. LINC01638 has been shown to interact with c-Myc and protect it from SPOP-mediated ubiquitination and degradation [97] (Figure 4). Reduced lncRNA levels of FGF13-AS1 have been detected in breast cancer cells and highly metastatic breast cancer cell lines. FGF13-AS1 inhibits tumor cell proliferation, migration, and invasion. This lncRNA binds specifically to the IGF2BP family of proteins and disrupts the interaction between IGF2BP and MYC mRNA. It leads to a decrease in the lifetime of MYC mRNA and, consequently, a lower level of the corresponding factor. Importantly, the c-Myc factor itself suppresses the expression of FGF13-AS1 [222] (Figure 4). Thus, any suppression of the expression or activity of this transcription factor can activate the FGF13-AS1 lncRNA-mediated regulatory mechanism, enhancing the suppression of MYC gene expression.

### 3.6. lncRNAs That Control MYC Gene Expression in Lung Cancer Cells

Several new lncRNAs affecting c-Myc factor expression have been found in lung cancer cells (Table 2). LINC01123 lncRNA in lung cancer cells forms a positive LINC01123/miR-199a-5p/MYC regulatory loop with c-Myc factor (Figure 4). Such regulatory loops may be a prospective target for therapeutic action, as suppression of the expression of this lncRNA inhibits the ability of cancer cells to proliferate [185]. An alternative isoform of the previously described PVT1 lncRNA, PVT1b, was also found in lung cancer cells. This isoform is synthesized under the influence of tumor suppressor p53 and, unlike the PVT1a isoform described above, suppresses the expression of the MYC gene (Figure 4). Increased expression of the PVT1b isoform in cancer cells slows down tumor growth [245].

### 3.7. lncRNAs Controlling MYC Gene Expression in Myeloma Cells

One important role of c-Myc factor in oncogenesis, as was mentioned earlier, is the formation of drug resistance in tumor cells. The role of PDIA3P lncRNA in this process has been demonstrated in multiple myeloma cells. This lncRNA interacts with the c-Myc factor and enhances its stimulatory effect on the glucose-6-phosphate dehydrogenase gene promoter (Figure 4), high levels of which allow for reducing the toxic effect of bortezomib on myeloma cells [56].

### 3.8. lncRNAs Controlling MYC Gene Expression in Medulloblastoma Cells

In nervous system tumors, gliomas and medulloblastomas, the regulation of MYC gene expression was found to be impaired by the binding of miR-33b-5p ceRNA DANCR (Figure 4). Suppression of this ceRNA’s expression leads to decreased levels of c-Myc factor and slows down cancer cell proliferation [171].

## 4. Circular RNA

A new type of RNA, circular RNA (circRNA), has been discovered relatively recently. This RNA type is characterized by a closed-loop structure and is, therefore, more resistant against the action of nucleases than linear RNA molecules. CircRNA is formed by splicing, so the same gene can be transcribed to both linear and circular RNA molecules. Due to the absence of a 5’-end and hence no cap structure, most circRNAs in eukaryotes are non-coding. However, circRNAs can perform a number of functions described for lncRNAs: they bind miRNAs, interact with regulatory sequences of the genome, and bind to proteins, altering their functions [265,266].

The role of circRNA in the development of various types of tumors has not been studied as well as for lncRNA and miRNA (Table 1, Table 2 and Table 3). In this section, we focus on the variety of mechanisms by which they affect c-Myc factor formation, function, and degradation. Some circRNAs bind miRNAs in different types of tumors. For example, increased expression of the cyclic isoform of the aforementioned PVT1 RNA, circPVT1, has been observed in leukemia, gastric, and colon cancer cells. This circRNA can activate MYC gene expression by binding miR-125 and miR-145 (Figure 5). Increased circPVT1 levels are associated with accelerated proliferation and increased tumor cell viability [267]. Similarly, circRNA_103809 enhances the ability of lung cancer cells to invasively grow by binding miR-4302, which suppresses ZNF121-dependent expression of the MYC gene [88] (Figure 5). For another RNA, circCCDC66, the ability to up-regulate MYC gene expression in colon cancer cells through the binding of miR-33b and miR-93 was shown (Figure 5). High levels of this circCCDC66 promote tumor growth and metastasis [155]. Additionally, high levels of this circRNA promote the development of gastric cancer [268]. Two other circRNAs, circLMTK2 and circ-PRMT5, have been shown to bind miR-150-5p, miR-145, and miR-1304 and increase MYC gene expression in gastric cancer cells (Figure 5). Suppression of the expression of these circRNAs reduces the proliferation and migration of tumor cells [166,167] A circRNA, circ_0068307, was also found to stimulate MYC gene expression and bladder cancer cell proliferation by binding miR-147 [150].

In some cases, circRNAs form regulatory loops with the MYC gene. For example, in gastric cancer cells, circ-NOTCH1 RNA is involved in the regulation of MYC gene expression. The expression of this circRNA is stimulated by c-Myc factor, while the RNA itself stabilizes MYC gene mRNA by binding miRNA miR-449c-5p [168] (Figure 5). Similarly, in oral squamous cell carcinoma cells, c-Myc factor activates the expression of circUHRF1 RNA, which in turn binds miR-526b, increasing the stability of this factor’s mRNA [191].

Some studies are able to trace longer chains of interactions linking circRNA and c-Myc factor activity. Thus, in gastric cancer cells, it was shown that the RNA circHECTD1 binds miR-1256, thus activating expression of the USP5 gene which in turn leads to stabilization of β-catenin which activates expression of the MYC gene. Another RNA affecting β-catenin activity is circRNA_102171. This RNA binds to the β-catenin-interacting protein CTNNBIP1, resulting in increased β-catenin activity and MYC gene expression in thyroid cancer cells (Figure 5). High levels of this circRNA stimulate tumor growth and the process of metastasis formation [277]. One more RNA which suppresses β-catenin activity in thyroid tumor cells is circ-ITCH. This circRNA binds miR-22-3p and increases the expression of CBL ubiquitin ligase, which suppresses β-catenin activity. Increased levels of circ-ITCH also suppress tumor growth and metastasis [276].

In addition to miRNA binding, circ-ITCH RNAs can influence MYC expression through direct interaction with the gene promoter. In colon cancer cells, circCTIC1 RNA binds BPTF and attracts it to the MYC gene promoter (Figure 5). High levels of this circRNA enhance MYC gene transcription and cancer cell proliferation [272]. In contrast, another circRNA, circNR3C1, inhibits the interaction of the BRD4 protein with the MYC gene promoter and suppresses the expression of this gene and bladder tumor cell proliferation [270].

CircRNA is also able to influence the stability and activity of the c-Myc factor itself. For example, the circRNA angiomotin-like1 (circ-Amotl1) binds to c-Myc factor and promotes its stabilization and transport to the nucleus. Increased expression of this RNA in breast cancer cells enhances tumor growth [271]. Another RNA, circECE1, also interacts with c-Myc protein and inhibits its ubiquitination and degradation (Figure 5). Its increased level is associated with activation of osteosarcoma cell proliferation and migration processes, as well as increased oxygen-free metabolism [275]. CircRNAs have also been found to decrease the stability of the c-Myc factor. CircCDYL RNA does not affect the mRNA level of the MYC gene, but it decreases the level of the corresponding protein, apparently decreasing its stability. High levels of this circRNA suppress bladder cancer cell proliferation and migration [269].

In rare cases, circRNA can work by means of encoded polypeptides. For example, circ-FBXW7 encodes the FBXW7-185 protein that binds to ubiquitin-specific peptidase 28 (USP28 protein). This protein interaction causes accelerated degradation of the peptidase and disrupts the stabilization of the c-Myc factor by this enzyme (Figure 5). Increased expression of circ-FBXW7 in glioblastoma cells suppresses their proliferation [274].

## 5. Conclusions

Numerous data in this review demonstrate that c-Myc plays an important role in the development of a wide variety of cancers. According to current reports, MYC expression levels are elevated in approximately 70% of human tumors [278,279]. However, there are still no drugs widely available in clinical practice which aim at suppressing the expression or activity of this oncogene [280]. Typically, low-molecular-weight compounds that specifically block the activity of the target protein are developed to block oncogene activity. However, the c-Myc molecule is not an enzyme and lacks the “pockets” to which low-molecular-weight inhibitors are usually matched [280,281]. To date, several molecules have been found that disrupt the binding of c-Myc and its partner Max, and stimulate c-Myc degradation by facilitating its phosphorylation by threonine 58 and subsequent ubiquitination [282]. A study of these molecules in animal models showed that the use of these inhibitors resulted in the enrichment of the tumor with cells with high PD-L1 expression, indicating the need for simultaneous use of c-Myc and PD-1 inhibitors [282]. Due to the high rate of change in c-Myc levels, prolonged and continuous exposure to these transcriptional factor inhibitors is required to effectively suppress its expression. The narrow therapeutic window of currently developed drugs makes it difficult to use them for tumor therapy [280].

An alternative approach to suppress c-Myc levels is the use of siRNA analogs. Two studies of such molecules currently exist, but both have been halted due to sponsor rejection (NCT02110563; NCT0231405). The use of RNA- and DNA-based drugs has been underdeveloped until recently due to low stability, difficulties in targeted delivery, and possible side effects [283]. However, the widespread use of RNA- and DNA-based vaccines against SARS-CoV2 could significantly advance the use of RNA- and DNA-containing drugs. The use of nuclease-protected siRNA analogs and single-stranded DNA complementary to target RNA may be an effective way to reduce the expression of certain genes in the long term [284]. However, in the case of the MYC gene, the question of the optimal sequence selection for the annealing therapeutic molecule arises. Known natural miRNAs that inhibit c-Myc synthesis may have additional targets, which may vary for different cell types. In addition, the set of lncRNAs and circRNAs capable of blocking certain miRNAs may differ in different cell types. When selecting a target, it is also important to consider positive and negative regulatory loops. As one of the solutions, a more long-lasting effect can be achieved by simultaneously blocking several RNAs involved in different regulatory loops.

This review describes different types of RNA that control MYC gene expression in different tissues and tumor types (Table 1, Table 2 and Table 3; Figure 6). Especially noteworthy is the diversity of different types of RNAs controlling the expression of this proto-oncogene in cells of digestive system cancers (Table 1, Table 2 and Table 3; Figure 7). Using tissue-specific regulatory RNAs rather than MYC gene mRNA as targets provides a potential opportunity to selectively influence c-Myc expression in cells of a particular tumor type. This may allow for creating a drug with a more selective effect and, consequently, a wider therapeutic window. It is worth noting that the studies on the role of different RNAs in the regulation of MYC expression in different cell types are not exhaustive, and some of the mentioned RNAs may function in a wider range of tissues and tumors than is currently known. 

Thus, the information provided in this review indicates the possibility of developing a specific diagnosis and treatment for different tumor types. Since suppression of MYC expression can reduce cell resistance to chemotherapy and radiotherapy, the use of tumor-specific MYC inhibitors can be applied to create effective anti-tumor therapy options.

## Figures and Tables

**Figure 1 biomedicines-09-00921-f001:**
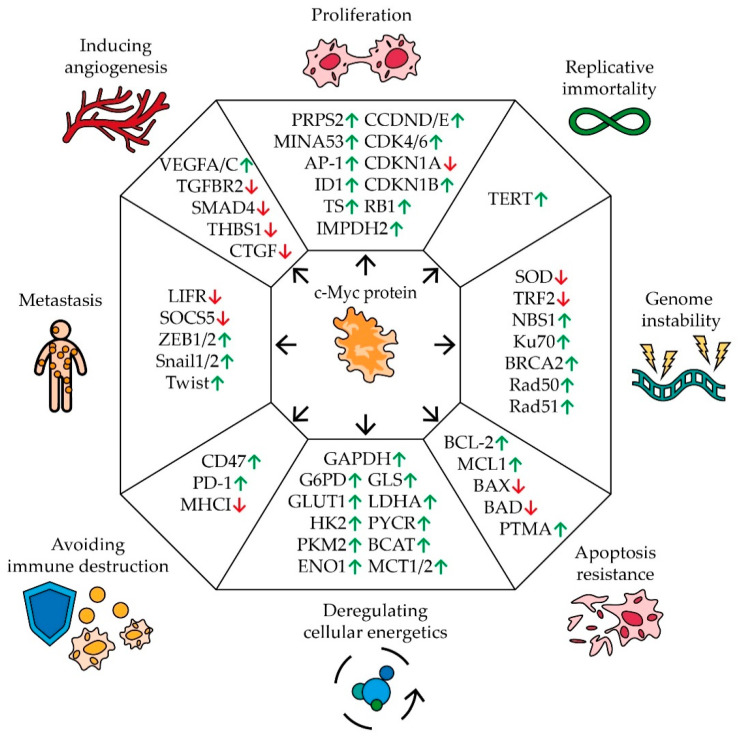
The impact of MYC proto-oncogene on the hallmarks of cancer development. Schematic representation of c-Myc’s effect on pivotal genes involved in carcinogenic pathways. Arrows indicate an increase (green) or a decrease (red) in gene expression in response to MYC expression (see text for description and references).

**Figure 2 biomedicines-09-00921-f002:**
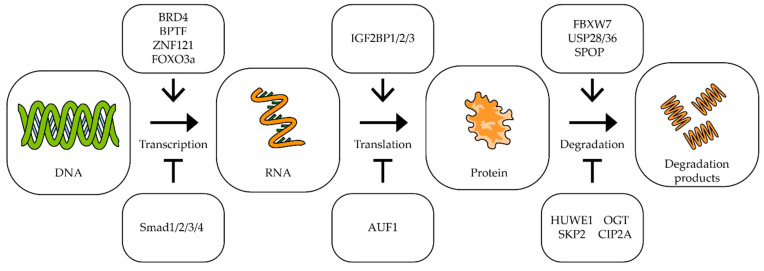
The impact of different factors on MYC transcription, translation and degradation. Schematic representation of influence of different factors on MYC transcription, translation, and c-Myc degradation. The upper/lower boxes indicate factors that activate/inhibit transcription, translation, or protein degradation (see text for description and references).

**Figure 3 biomedicines-09-00921-f003:**
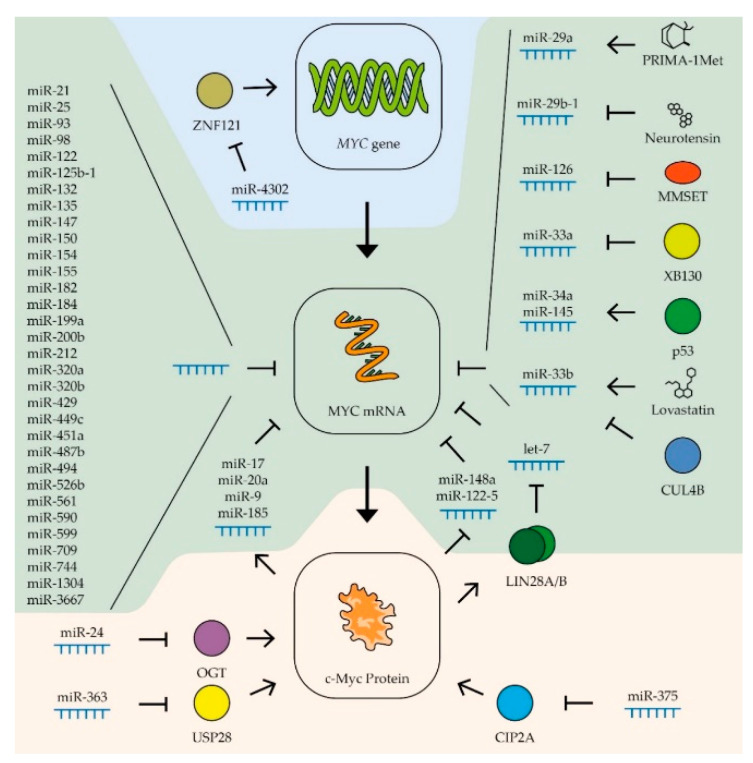
Control of MYC expression by miRNAs. Schematic representation of miRNAs’ influence on MYC gene transcription (blue zone), MYC mRNA (green zone), and c-Myc protein stability (yellow zone). Colored circles indicate the proteins included in the pathway. (See text for description and references).

**Figure 4 biomedicines-09-00921-f004:**
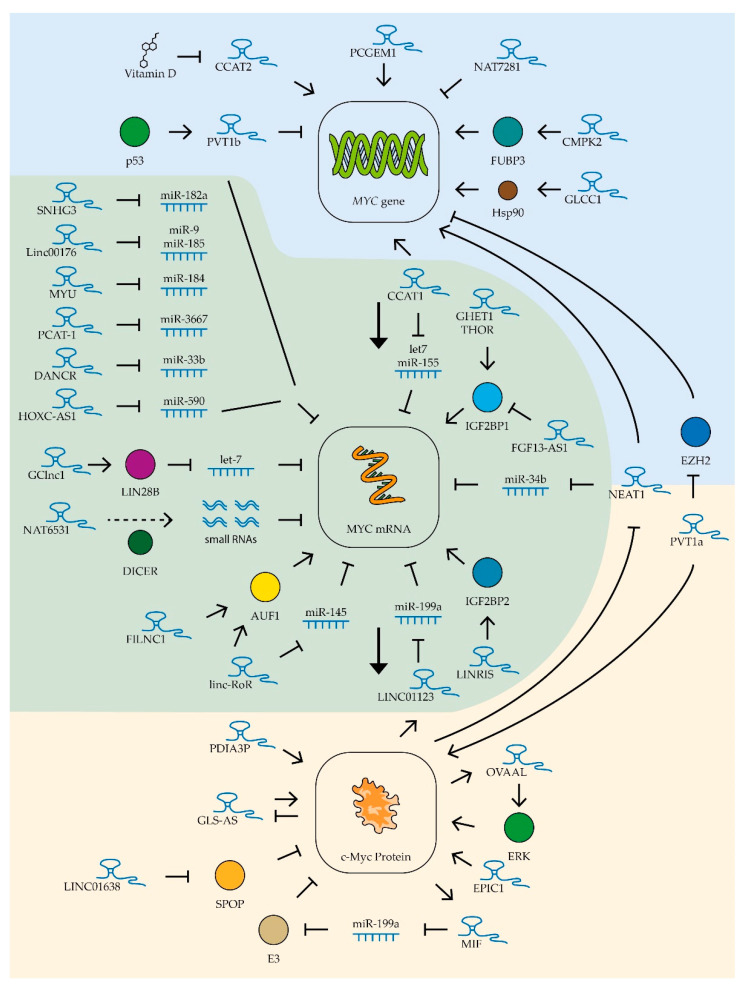
Control of MYC expression by lncRNAs. Schematic representation of lncRNAs’ influence on MYC gene transcription (blue zone), MYC mRNA (green zone), and c-Myc protein stability (yellow zone). The colored circles indicate the proteins included in the pathway, the dotted line is product processing. See text for description and references.

**Figure 5 biomedicines-09-00921-f005:**
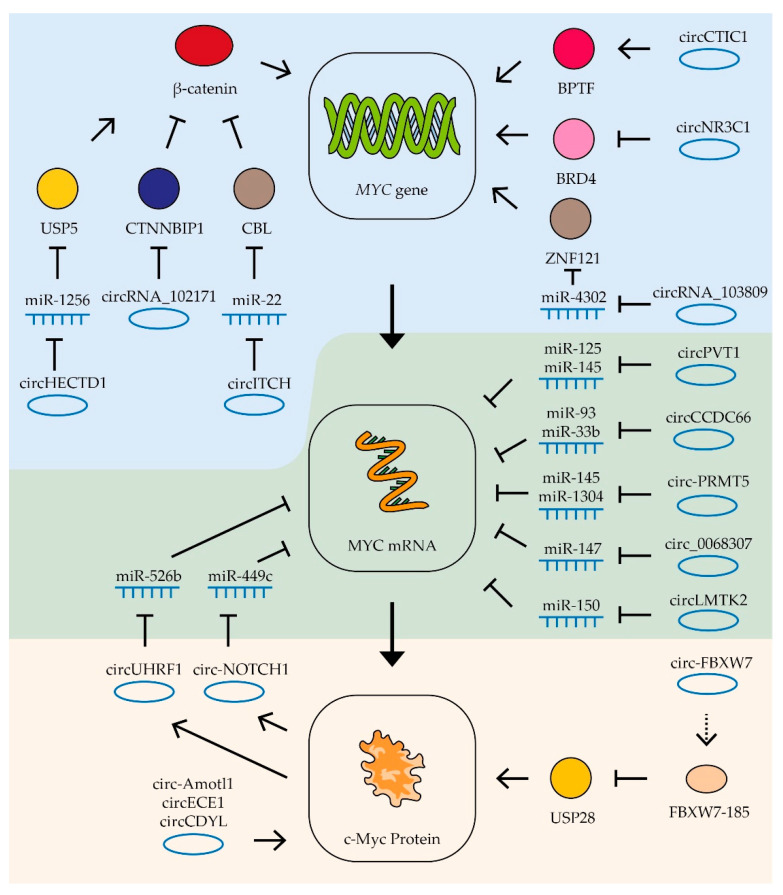
Control of MYC mRNA by circRNAs. Schematic representation of lncRNAs’ influence on MYC gene transcription (blue zone), MYC mRNA (green zone), and c-Myc protein stability (yellow zone). Colored circles indicate the proteins included in the pathway, the dotted line is a translation product. See text for description and references.

**Figure 6 biomedicines-09-00921-f006:**
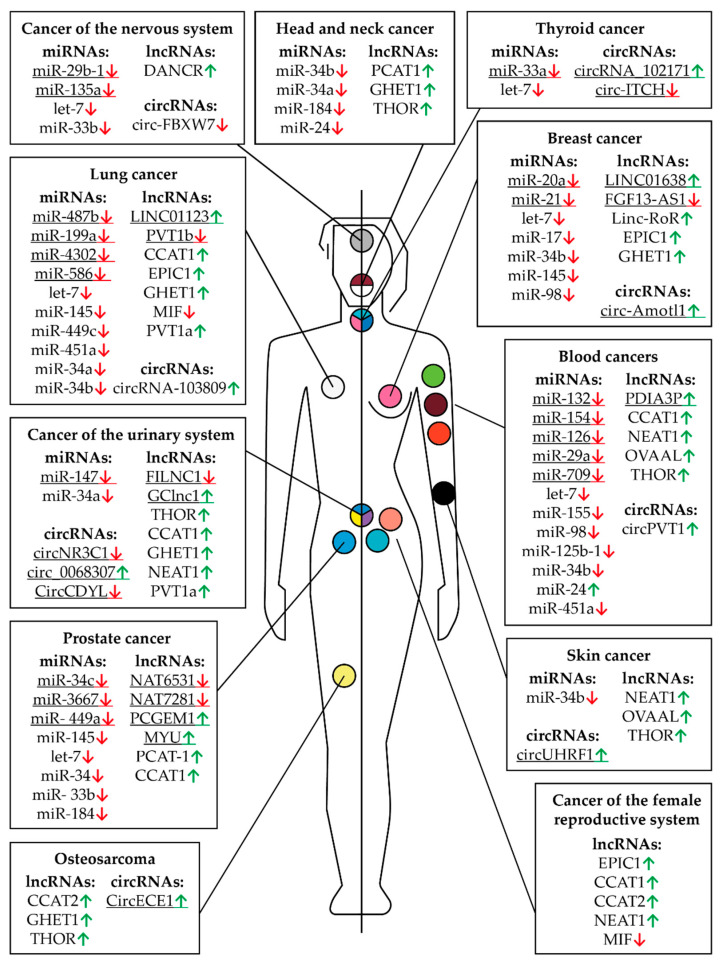
Controlling MYC RNAs in different types of cancer except those of the digestive system. The fields indicate miRNAs, lncRNAs, and circRNAs that have altered expression compared to healthy tissues in a particular cancer. Arrows indicate an increased (green) or decreased (red) level of RNA in a particular type of cancer. The human body (the left part is a male, the right part is female) and the location of cancer tumors are presented schematically. The colored circles indicate a particular type of cancer: burgundy—multiple myeloma, orange—leukemia, pink—breast cancer, peach—uterine or endometrial cancer, yellow—sarcoma or bone cancer, lime green—non-Hodgkin lymphoma, teal—ovarian cancer, light blue—prostate cancer, black—skin cancer, gray—brain cancer, white—lung cancer, blue, yellow, and purple—bladder cancer, blue, pink, and teal—thyroid cancer, white and burgundy—head and neck cancer. The underlined RNAs are currently believed to be tissue specific but new roles can potentially be discovered.

**Figure 7 biomedicines-09-00921-f007:**
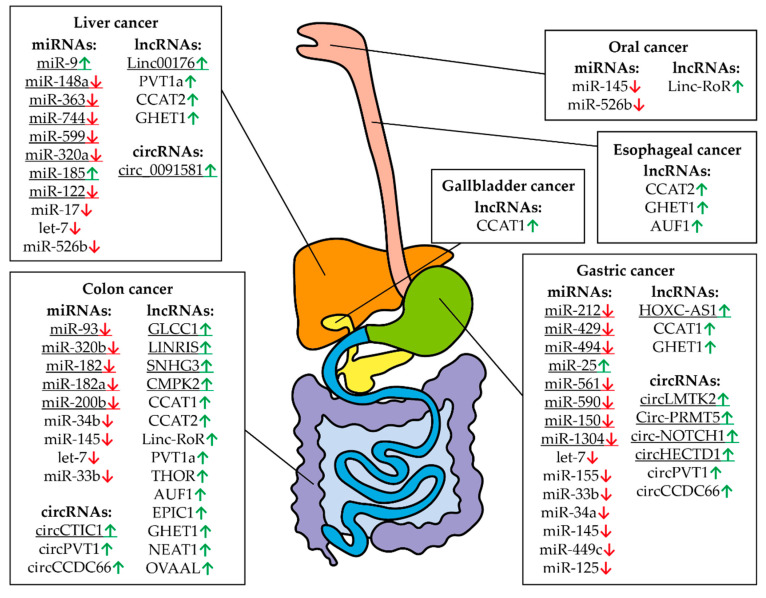
Controlling MYC RNAs in different types of digestive system cancers. The fields indicate miRNAs, lncRNAs, and circRNAs that have altered expression compared to healthy tissues in a particular cancer. Arrows indicate an increased (green) or decreased (red) level of RNA in a particular type of cancer. The digestive system is represented schematically. The underlined RNAs are currently believed to be tissue specific but new roles can potentially be discovered.

**Table 1 biomedicines-09-00921-t001:** miRNAs that control the expression of the MYC gene in tumors of various human organs.

Cancer	MiRNA	Alteration in Cancer	Mechanistically {Association with the Altered Level of this RNA in Tumor Cells}	References
Acute myeloid leukemia	let-7	Down-regulated	{Poor prognosis}	[130]
	miR-155	Down-regulated	Inhibits cancer cell proliferation	[149]
Bladder cancer	miR-147	Down-regulated	Inhibits cancer cell proliferation	[150]
Burkitt lymphoma	let-7a let-7b miR-98	Down-regulated	Reverses MYC-induced growth	[120]
	miR-34b-5p let-7c	Down-regulated	G1 arrest	[151]
	let-7-5p miR-132-5p miR-125b-1 miR-154	Down-regulated	Inhibits cancer cell proliferation	[121]
Breast cancer	miR-17-5p miR-20a-5p	Down-regulated	Inhibits cancer cell proliferation	[152]
	miR-34b-3p	Down-regulated	{Associated with metastasis development}	[153]
	miR-145-5p	Down-regulated	{Poor prognosis}	[154]
	miR-21-5p miR-98-5p let-7	Down-regulated	Suppresses cancer cell growth	[136]
	let-7	Down-regulated	{Poor prognosis}	[131]
	let-7	Down-regulated	Inhibits cancer cells proliferation	[137,138]
Colon cancer	miR-34b-3p	Down-regulated	{Associated with metastasis development}	[153]
	miR-145-5p	Down-regulated	{Poor prognosis}	[154]
	let-7	Down-regulated	Suppresses cancer cell growth	[142,143]
	miR-33b miR-93	Down-regulated	Inhibited cell proliferation, migration, and invasion	[155]
Colorectal cancer	miR-320b	Down-regulated	Inhibits cancer cell proliferation	[156]
	miR-182-5p	Down-regulated	Inhibits cancer cell proliferation	[156]
	miR-182a-5p	Down-regulated	Inhibits cancer cell proliferation	[157]
	miR-200b-3p	Down-regulated	Inhibits cancer cell proliferation	[158]
Diffuse large B-cell lymphoma	miR-34b-5p	Down-regulated	Inhibits cancer cell proliferation	[159]
Gastric cancer	miR-212-3p	Down-regulated	Inhibits cancer cell proliferation	[160]
	miR-429	Down-regulated	Inhibits cancer cell viability, proliferation, and attachment	[123]
	let-7	Down-regulated	{Poor prognosis}	[132]
	miR-494-3p	Down-regulated	Inhibits cancer cells proliferation	[145]
	miR-155-5p	Down-regulated	Inhibits cancer cell growth and invasion	[144]
	miR-33b-5p	Down-regulated	Inhibited cell proliferation, migration, and invasion	[161]
	miR-25-5p	Up-regulated	Inhibits cancer cell apoptosis	[162]
	miR-34a	Down-regulated	Inhibits cancer cell growth and invasion	[163]
	miR-561-3p	Down-regulated	Inhibits cancer cell growth and invasion	[164]
	miR-590-3p	Down-regulated	Inhibits cancer cell proliferation	[165]
	miR-150-5p	Down-regulated	Inhibits cell proliferation and migration	[166]
	miR-145 miR-1304	Down-regulated	Inhibits cell proliferation	[167]
	miR-449c-5p	Down-regulated	Inhibits cell proliferation and migration	[168]
	miR-125	Down-regulated	Inhibits cell proliferation	[169]
Glioma	miR-29b-1	Down-regulated	Inhibits cancer cell proliferation	[170]
	miR-33b-5p	Down-regulated	Inhibits cancer cell proliferation	[171]
	miR-135a-5p	Down-regulated	Inhibits cancer cell proliferation	[172]
Head and neck carcinoma	miR-34b-3p	Down-regulated	{Associated with metastasis development}	[153]
	miR-34a-5p	Down-regulated	Attenuates tumor growth and metastasis	[173]
Hodgkin Lymphoma	miR-24-3p	Up-regulated	Protects cancer cells from apoptosis	[124]
Liver cancer	let-7g	Down-regulated	Inhibits proliferation of hepatocellular carcinoma cells	[139]
	let-7	Down-regulated	{Poor prognosis}	[133]
	miR-148a-5p miR-363-3p	Down-regulated	G1 arrest	[174]
	miR-744-5p	Down-regulated	Inhibits cancer cell proliferation	[175]
Liver cancer	miR-599	Down-regulated	Inhibits cancer cell proliferation, migration, and invasion	[176]
	miR-320a	Down-regulated	Inhibits tumor proliferation and invasion	[177]
	let-7	Down-regulated	Inhibits cancer cell proliferation	[140]
	miR-9 miR-185-5p	Up-regulated	Inhibits cancer cell proliferation and survival	[178]
	miR-17-5p	Down-regulated	Represses invasiveness and metastasis, increases survival	[127]
	miR-122-5p	Down-regulated	Inhibits cancer cell proliferation	[179]
	miR-526b	Down-regulated	Inhibits cancer cell proliferation	[180]
Lung cancer	miR-34b-3p	Down-regulated	{Associated with metastasis development}	[153]
	let-7a-5p	Down-regulated	Inhibits the growth of lung cancer	[141]
	miR-145-5p	Down-regulated	Inhibits cancer cell proliferation	[181]
	miR-487b-3p	Down-regulated	Inhibits cancer cell growth and invasion	[182]
	miR-449c-5p	Down-regulated	Inhibits cancer cell proliferation	[183]
	miR-451a	Down-regulated	Reverses EMT to mesenchymal–epithelial transition	[146]
	miR-34a-5p	Down-regulated	Inhibits cancer cell proliferation	[184]
	miR-199a-5p	Down-regulated	Inhibits cancer cell proliferation	[185]
	miR-4302	Down-regulated	Inhibits cancer cell proliferation and invasion	[88]
	miR-586-5p	Up-regulated	Enhances cancer cell proliferation	[59]
Medulloblastoma	miR-33b-5p	Down-regulated	Inhibits cancer cell proliferation	[186]
Melanomas	miR-34b-3p	Down-regulated	{Associated with metastasis development}	[153]
Myeloma	miR-126-5p	Down-regulated	Inhibits cancer cell proliferation	[187]
	miR-29a-3p	Down-regulated	Inhibits cancer cell viability	
Nasopharyngeal carcinoma	miR-184	Down-regulated	Blocks cell growth and survival	[188]
	miR-24-3p	Down-regulated	Suppresses metastasis	[125]
Neuroblastoma	let-7	Down-regulated	{Worse overall survival}	[134]
Oral squamous cell carcinoma	miR-145-5p	Down-regulated	Inhibits cancer cell proliferation	[189,190]
	miR-526b-5p	Down-regulated	Inhibits cancer cell proliferation	[191]
Prostate cancer	miR-34c-5p	Down-regulated	Inhibits cancer cell proliferation	[192]
	miR-145-5p	Down-regulated	Inhibits cancer cell proliferation	[193]
	let-7	Down-regulated	Higher in non-metastatic tumors	[135]
	miR-34 let-7	Down-regulated	{Poor prognosis}	[194]
	miR-3667-3p	Down-regulated	Inhibits cancer cell proliferation	[195]
	miR- 449a	Down-regulated	Enhances cancer cell radiosensitivity	[196]
	miR- 33b	Down-regulated	Inhibits cancer cell proliferation	[197]
	miR-184	Down-regulated	Inhibits cancer cell proliferation	[198]
Renal cell carcinoma	miR-34a-5p	Down-regulated	Suppresses malignant transformation	[199]
T-cell acute lymphoblastic leukemia	miR-451a miR-709	Down-regulated	Inhibits cancer cell proliferation	[200]
Thyroid cancer	let-7f-5p	Down-regulated	Inhibits cancer cell proliferation	[201]
	miR-33a-5p	Down-regulated	Inhibits cancer cell proliferation	[202]

**Table 2 biomedicines-09-00921-t002:** lncRNAs that control the expression of the MYC gene in tumors of various human organs.

Cancer	lncRNA	Alteration in Cancer	Mechanistically {Association with the Altered Level of this RNA in Tumor Cells}	References
Acute myeloid leukemia	CCAT1	Up-regulated	Promotes cancer cell proliferation and survival	[149]
Bladder cancer	GClnc1	Up-regulated	Promotes cancer cell proliferation, metastasis, and invasiveness	[217]
	GHET1	Up-regulated	{Predicts an unfavorable survival}	[218]
	NEAT1	Up-regulated	Promotes cancer cell proliferation, invasion, and survival	[219]
	PVT1a	Up-regulated	Promotes cancer cell proliferation and invasion	[220]
Breast cancer	EPIC1	Up-regulated	Promotes cancer cell proliferation and survival	[221]
	GHET1	Up-regulated	{Predicts an unfavorable survival}	[218]
	FGF13-AS1	Down-regulated	Suppresses cancer cell proliferation, migration, and invasion	[222]
	LINC01638	Up-regulated	{Predicts a poor outcome}	[97]
	Linc-RoR	Up-regulated	Suppresses cancer cell proliferation	[223]
Cervical cancer	MIF	Down-regulated	Suppresses cancer cell proliferation	[59]
Cholangiocarcinoma	EPIC1	Up-regulated	Promotes cancer cell proliferation	[224]
Chronic myeloid leukemia	NEAT1	Up-regulated	Promotes cancer cell proliferation and survival	[225]
Colon cancer	CCAT1	Up-regulated	Promotes cancer cell proliferation, migration, and invasion	[213]
	CCAT2	Up-regulated	Promotes metastatic progression and chromosomal instability in colon cancer	[226]
	Linc-RoR	Up-regulated	Promotes cancer cell proliferation	[223]
	PVT1a	Up-regulated	Promotes cancer cell proliferation and invasion	[227]
	THOR	Up-regulated	Promotes cancer cell proliferation and migration	[228]
	AUF1	Up-regulated	Promotes cancer cell proliferation	[223]
	EPIC1	Up-regulated	Promotes cancer cell proliferation and invasion	[229]
Colorectal cancer	GHET1	Up-regulated	Promotes cancer cell proliferation	[230]
	GLCC1	Up-regulated	Promotes cancer cell survival and proliferation	[231]
	LINRIS	Up-regulated	Promotes cancer cell proliferation	[232]
	NEAT1	Up-regulated	Promotes cancer cell proliferation and survival	[233,234]
	SNHG3	Up-regulated	Promotes cancer cell proliferation	[157]
	OVAAL	Up-regulated	Promotes cancer cell proliferation	[206]
	CMPK2	Up-regulated	Promotes colorectal cancer progression	[235]
Diffuse large B-cell lymphoma	NEAT1	Up-regulated	Promotes cancer cell proliferation and survival	[159]
Endometrial adenocarcinoma	NEAT1	Up-regulated	Promotes cancer cell proliferation, invasion, and migration	[236]
Gallbladder cancer	CCAT1	Up-regulated	Promotes cancer cell proliferation and survival	[214]
Gastric cancer	CCAT1	Up-regulated	Promotes cancer cell proliferation, migration, and invasion	[237]
	GHET1	Up-regulated	Promotes cancer cell proliferation	[238]
	HOXC-AS1	Up-regulated	Promotes cancer cell proliferation and metastasis	[165]
Glioma	DANCR	Up-regulated	Promotes cancer cell proliferation	[171]
Head and neck cancer	GHET1	Up-regulated	{Predicts an unfavorable survival}	[218]
	PCAT-1	Up-regulated	Promotes cancer cell proliferation	[239]
Hepatocellular carcinoma	PVT1a	Up-regulated	Promotes cancer cell proliferation and invasion	[240]
Liver cancer	CCAT2	Up-regulated	Promotes cancer cell proliferation and invasion	[241]
	GHET1	Up-regulated	{Predicts an unfavorable survival}	[218]
	Linc00176	Up-regulated	Promotes cancer cell proliferation	[178]
Lung cancer	CCAT1	Up-regulated	Promotes cancer cell proliferation and survival	[209,242]
	EPIC1	Up-regulated	Promotes cancer cell proliferation	[243]
	GHET1	Up-regulated	{Predicts an unfavorable survival}	[218]
	LINC01123	Up-regulated	Promotes cancer cell proliferation	[185]
	MIF	Down-regulated	Suppresses cancer cell proliferation	[59]
	PVT1a	Up-regulated	Promotes cancer cell proliferation and invasion	[244]
	PVT1b	Down-regulated	Suppresses cancer cell proliferation	[245]
Medulloblastoma	DANCR	Up-regulated	Promotes cancer cell proliferation	[171]
Melanoma	OVAAL	Up-regulated	Promotes cancer cell proliferation	[206]
	THOR	Up-regulated	Promotes cancer cell proliferation	[246]
Multiple myeloma	PDIA3P	Up-regulated	Enhances cancer cell proliferation and drug resistance	[56]
Nasopharyngeal carcinoma	THOR	Up-regulated	Promotes cancer cell proliferation	[247]
Esophageal cancer	CCAT2	Up-regulated	Promotes radiotherapy resistance	[248]
	GHET1	Up-regulated	{Predicts an unfavorable survival}	[218]
	AUF1	Up-regulated	Promotes cancer cell proliferation	[249]
Oral cancer	Linc-RoR	Up-regulated	{Associated with tumor recurrence and poor therapeutic response}	[190]
Osteosarcoma	CCAT2	Up-regulated	Promotes cancer cell proliferation and invasion	[250]
	GHET1	Up-regulated	{Predicts an unfavorable survival}	[218]
	THOR	Up-regulated	Promotes cancer cell proliferation	[251]
Ovarian cancer	EPIC1	Up-regulated	Promotes cancer cell proliferation and survival	[252]
	CCAT1	Up-regulated	Promotes cancer cell proliferation and survival	[215]
	CCAT2	Up-regulated	Promotes cancer cell proliferation and invasion	[253]
Pancreatic cancer	GHET1	Up-regulated	{Predicts an unfavorable survival}	[218]
	GLS-AS	Down-regulated	Suppresses cancer cell proliferation	[207]
Prostate cancer	CCAT1	Up-regulated	Promotes cancer cell proliferation and survival	[211]
	NAT6531 NAT7281	Down-regulated		[254]
	PCAT-1	Up-regulated	Promotes cancer cell proliferation	[195]
	PCGEM1	Up-regulated	Promotes cancer cell proliferation and survival	[255]
	MYU	Up-regulated	Promotes cancer cell proliferation	[198]
Renal cancer	CCAT1	Up-regulated	Promotes cancer cell proliferation and survival	[212]
	FILNC1	Down-regulated	Inhibits tumor development	[256]
Renal cell carcinoma	THOR	Up-regulated	Promotes cancer cell proliferation	[257]
Retinoblastoma	THOR	Up-regulated	Promotes cancer cell proliferation	[258]
Squamous cell carcinoma	NEAT1	Up-regulated	{Worse overall survival}	[259]
Uterine cervical cancer	CCAT2	Up-regulated	Progression of uterine cervical cancer	[260]

**Table 3 biomedicines-09-00921-t003:** CircRNAs that control the expression of the MYC gene in tumors of various human organs.

Cancer	circRNA	Alteration in Cancer	Mechanistically	References
Bladder cancer	CircCDYL	Down-regulated	Suppresses cell growth and migration	[269]
	circNR3C1	Down-regulated	Suppresses cancer cell growth	[270]
	circ_0068307	Up-regulated	Promotes cancer cell migration and proliferation	[150]
Breast cancer	circ-Amotl1	Up-regulated	Stimulates tumor growth	[271]
Colon cancer	circPVT1	Up-regulated	Promotes cancer cell proliferation	[267]
	circCTIC1	Up-regulated	Promotes cancer cell proliferation	[272]
	circCCDC66	Up-regulated	Promotes cancer cell proliferation and migration	[155]
Gastric cancer	circLMTK2	Up-regulated	Promotes cancer cell proliferation	[166]
	Circ-PRMT5	Up-regulated	Promotes cancer cell proliferation	[167]
	circ-NOTCH1	Up-regulated	Promotes cancer cell migration and proliferation	[168]
	circPVT1	Up-regulated	Promotes cancer cell proliferation	[267]
	circHECTD1	Up-regulated	Promotes cancer cell proliferation	[273]
	circCCDC66	Up-regulated	Promotes cancer cell proliferation	[268]
Glioblastoma	circ-FBXW7	Down-regulated	Suppresses cancer cell growth	[274]
Leukemia	circPVT1	Up-regulated	Promotes cancer cell proliferation	[267]
Liver cancer	circ_0091581	Up-regulated	Stimulates tumor growth	[180]
Lung cancer	circRNA_103809	Up-regulated	Stimulates cancer cell proliferation and invasion	[88]
Osteosarcoma	CircECE1	Up-regulated	Promotes cancer cell migration and proliferation	[275]
Squamous cell carcinoma	circUHRF1	Up-regulated	Promotes cancer cell proliferation	[191]
Thyroid cancer	circ-ITCH	Down-regulated	Suppresses cancer cell migration and proliferation	[276]
	circRNA_102171	Up-regulated	Promotes cancer cell migration and proliferation	[277]

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
