# Peer review of "The Role of Non-Coding RNAs in the Regulation of the Proto-Oncogene MYC in Different Types of Cancer"

_biomedicines, 2021, doi:10.3390/biomedicines9080921_

Round 1
Reviewer 1 Report
The review is comprehensive with very detailed sections on the noncoding RNAs (miRNA, LncRNA, circRNA) and their interactions with MYC. As the authors are readily aware, MYC is involved in so many malignancies that it is almost the exception to find a disease that doesn't have MYC involvement. Therein lies my major concern with Figures 6 and 7 that depict controlling MYC RNAs in different types of cancer (Fig. 6) and different digestive system cancers (Fig. 7). It is unlikely that the miRNAs, lncRNAs and circRNAs listed for certain types is really specific to those types as depicted. There is already overlap with Pvt1a, for example, in lung, urinary and liver cancer, but these are only the systems that have been referenced. Many studies have shown involvement of PVT1(a) in virtually every type that has been listed in Figures 6 and 7 (the references are many) and so it is likely that a lincRNA or circRNA for Pvt1 is like involved in many more types than lung, urinary and liver. This was just an example, but the same can be applied to many other miRNAs, lncRNAs and circRNAs. I would suggest removing these figures as they are incomplete.
Author Response
We provide a point-by-point response to the reviewers’ comments.
Comment 1: Therein lies my major concern with Figures 6 and 7 that depict controlling MYC RNAs in different types of cancer (Fig. 6) and different digestive system cancers (Fig. 7). It is unlikely that the miRNAs, lncRNAs and circRNAs listed for certain types is really specific to those types as depicted.
Response: Certainly many of the non-coding RNAs affect Myc in various types of cancer. We reflected this in our manuscript and also depict it in Figures 6 and 7. According to the reviewer's comment, figures 6 and 7 are incomplete. As a solution, we have decided to highlight those RNAs for which the effect on other types of cancer is still unknown and now they can still be considered tissue-specific, and also we have added a comment in figure legend about potentially new roles of RNAs that can be discovered over time (Lines 754-755, 780-781).
We would like to thank the reviewers again for taking the time to review our manuscript.

Reviewer 2 Report
Reviewer’s Comment:
The present review “The role of non-coding RNAs in the regulation of the proto-oncogene MYC in different types of cancer” by Stasevich EM et al nicely demonstrated while covering different forms of ncRNAs controlling MYC expression causing various types of cancer in the human body. This is a comprehensive review mostly covering all the pertinent areas of MYC regulation except a couple of minor points stated below.
- The normal functional role of MYC proteins in the early stage of development needs to be mentioned a bit before moving into its potential role in cancer. A brief section regarding this perspective can be added on page2 after line 47.
- There are several typos, mismatched phrases in the manuscript which need to be corrected.
Author Response
We provide a point-by-point response to the reviewers’ comments.
• Comment 1: The normal functional role of MYC proteins in the early stage of development needs to be mentioned a bit before moving into its potential role in cancer. A brief section regarding this perspective can be added on page2 after line 47.
Response: Following this recommendation, we’ve added information regarding the functional role of MYC in the early stage of development (Lines 40-44).
• Comment 2: There are several typos, mismatched phrases in the manuscript which need to be corrected.
Some mistakes have been found and corrected (Lines 70, 108, 118, 142-143, 156, 181, 215, 234, 342, 383, 467, 472, 653, 657, 702, 745, 801,804).
We would like to thank the reviewers again for taking the time to review our manuscript.
